# Use of Antivibration Technology to Reduce Demands for In-Home Nursing Care and Support in Rural Settings for Persons with Essential Tremors: A Qualitative Study

**DOI:** 10.3390/ijerph21060714

**Published:** 2024-05-31

**Authors:** Fatemeh Mohammadnejad, Shannon Freeman, Tammy Klassen-Ross, Dawn Hemingway, Davina Banner

**Affiliations:** 1School of Health Sciences, University of Northern British Columbia, Prince George, BC V2N 4Z9, Canada; fatemeh.mohammadnejad@alumni.unbc.ca (F.M.); tammy.klassen-ross@unbc.ca (T.K.-R.); 2School of Nursing, University of Northern British Columbia, Prince George, BC V2N 4Z9, Canada; davina.banner-lukaris@unbc.ca; 3School of Social Work, University of Northern British Columbia, Prince George, BC V2N4Z9, Canada; dawn.hemingway@unbc.ca

**Keywords:** Parkinson’s disease, gerontechnology, aging, aging in place, interRAI, homecare, independent living

## Abstract

Introduction: With the increased integration of technologies in the healthcare sector, it is important to understand the benefits emerging technologies may play to reduce demands on the health care system. The Steadiwear antivibration glove shows promise for enhancing the independence in functional abilities for persons with essential tremors and for alleviating the need for support from the health care system. The objective of this study was to examine Registered Nurses’ (RN) perceptions of the potential for the Steadiwear antivibration glove to reduce the need for in-person support from community healthcare workers. Methods: Eleven RNs, experienced in providing care in rural communities, participated in a semi-structured interview sharing their perspectives towards use of the Steadiwear antivibration glove in community practice settings. Thematic analysis guided by Braun and Clarke was undertaken. Results: Nurses described the value of this technology to reduce client needs for support for activities of daily living (e.g., dressing, feeding) and independent activities of daily living (e.g., banking, transportation). Conclusions: Enhanced access to this technology may reduce the need for nursing and personal care support from the health system. Therefore the Steadiwear antivibration glove also shows potential to delay and/or prevent the need for more intensive support and mitigate the need for transition to a long-term care facility.

## 1. Introduction

Essential tremor, a neurological disorder characterized by involuntary and rhythmic shaking of the hands, head, or other body parts [1,2], is one of the most common movement disorders, affecting approximately 4% of the population over the age of 40 [3]. The global prevalence of essential tremor estimated to be 13.3 individuals per 1000 increases substantially to 57.9 individuals per 1000 persons over the age of 65 years [4]. Essential tremors can have a significant impact on a person’s daily life and may affect the ability to perform activities of daily living (ADL), such as eating, drinking, and dressing, as well as instrumental activities of daily living (IADL), such as driving, working, and managing finances. This can lead to decreased independence and increased reliance on others. Social well-being may also be impacted as persons living with essential tremors may experience embarrassment, anxiety, and depression due to the visible nature of the disorder [5,6,7]. The impacts on the daily life of the individual also affect family, friends, and healthcare providers who may need to provide increasing levels of care and support.

For individuals with essential tremor, the support of community healthcare providers, including Registered Nurses (RNs) and Care Aides, can be especially important. Without ample support, individuals with essential tremors may be at risk for premature entry to long-term care facilities [8]. Community healthcare providers play a pivotal role to assist individuals living with essential tremors who wish to age in place (remain at in their own home in community as they age). By providing care and support in the community, RNs can help individuals with essential tremors age in place and maintain their quality of life [9]. Registered Nurses promote health, prevent illness, and manage chronic conditions through ongoing assessment, monitoring, and care coordination. By providing services in the community, RNs help their clients with essential tremor to maintain as much independence as possible and remain connected to their families and social networks [10]. In rural areas, access to healthcare and community services can be challenging with limited ability to access support from formal care providers. Nursing workload, RN recruitment, and RN retention challenges enhanced in the rural context have been further compounded by staff burnout due to the COVID-19 pandemic.

The role of technology in the health sector has grown exponentially over the past few decades [11,12]. Technology affects health delivery and implementation; it has been shown to improve the quality of nurses’ decision making, and thereby also improve the quality of healthcare services [13], clinical improvement, and patients’ satisfaction [8]. Technology can provide individuals with supportive services, and it offers valuable services to patients and healthcare workers. Many technologies have been shown to reduce the workload of health care staff including online booking applications to make physician appointments more accessible, automated medication dispensers, and daily care reminders for self-management of health conditions [14,15]. Concurrently, technologies play an important role in supporting the user’s independence and ability to remain in their own homes. 

### Steadiwear Antivibration Glove

The Steadiwear antivibration glove is a new health technology designed to support enhanced independence in activities for persons living with essential tremor. The use of such technologies may be feasible in rural areas, where healthcare providers usually experience a higher workload in comparison to urban areas [16]. For persons with essential tremor, it is hypothesized that the Steadiwear antivibration glove may reduce reliance on support from RNs and care aides as the glove is noted to support the patients’ independence to perform Activities of Daily Living (ADL) and Instrumental Activities of Daily Living (IADL). 

Given the growing challenges with RN retention, recruitment, and support, especially in rural communities, it is important to understand the role that the Steadiwear antivibration glove may have to not only support persons living with essential tremors but also to reduce demands on healthcare staff workload. Therefore, the aim of this study was to describe the RN perspectives towards use of the Steadiwear antivibration glove to support persons with essential tremors living in rural communities and to examine the effects this may have on RN workload.

## 2. Methods

This study adopted a descriptive and inductive qualitative approach to explore Registered Nurses’ (RN) perceptions of the potential for the Steadiwear antivibration glove. This approach was selected as it enabled an in-depth analysis of experiences as they occur in the natural context.

### 2.1. Sampling and Recruitment

Recruitment to this descriptive exploratory qualitative study involved a purposive sample of 11 RNs. To be eligible to participate, participants needed to be RNs with experience working in a rural community (defined as a community with less than 10,000 residents) [17] and in community health settings, including home and community care, community case managers, and long-term care nursing. Posters were shared through existing networks and contacts of research team members DBL and SF by e-mail. Interested participants who saw the posters then independently emailed the research team indicating their interest to participate. Research team member FM followed up with potential participants sharing the study information letter and consent form. Participants were able to ask questions and all those who indicated interest to the research team agreed to participate. Furthermore, participants were also recruited through snowball sampling. Interviews were conducted and recorded over zoom in fall and winter 2021. Prior to data collection, all participants were provided with a comprehensive information sheet and were provided with opportunities to ask questions and seek clarification.

The initial aim was to recruit ten to fifteen RNs with experience working in rural communities; however, it was determined that data saturation was met after interviewing 11 participants as no new major themes emerged [18].

### 2.2. Data Collection and Analysis

Semi-structured interviews were performed to gather information on RNs perspectives towards technology use in general and more specifically towards the views on use of the Steadiwear antivibration glove for persons with essential tremor. First, participants were shown a video involving a person living with essential tremor performing various activities, including writing with a pen on a paper and drinking water from a glass, before and while using the Steadiwear antivibration glove (https://www.youtube.com/watch?v=5urBgJeiu84). This video also captured the person with essential tremor describing the impact use of the Steadiwear antivibration glove had for them. Following review of the video, participants participated in an interview, comprising open-ended and semi-structured questions. An interview guide was used to guide data collection, which evolved as the data collection and analysis unfolded. During the interviews, participants reflected hypothetically on the usability of the Steadiwear antivibration glove noting that the glove showed promise to enhance the user’s independence and reduce reliance of family and caregivers. Each interview lasted for approximately an hour in duration and was conducted remotely using a secure Zoom platform. Data were transcribed verbatim, and transcripts were reviewed for accuracy. 

A thematic analysis approach guided by Braun and Clarke [19] was used to analyze the study data. The following five-step analysis was implemented in the present study: (1) organizing and preparing the data for analysis, (2) reading or looking at all the data, (3) coding all the data, (4) generating descriptions and themes, and (5) representing the descriptions and themes [19]. The above-mentioned steps guided the generation of the data set. Following the five-step analysis enabled the researcher to generate in-depth and detailed descriptions of the analysis, further supporting transparency and reproducibility. Throughout the analysis, an inductive approach was used to generate descriptive themes (See Appendix A) and illustrative quotes were selected. Transcripts were reviewed in detail several times, allowing the researcher to gain familiarity and connection with the data. While reading and reviewing the transcriptions, important points were noted, data were coded, and segments of text were grouped and labelled. Findings were color-coded and linked to form themes. To ensure confidentiality of participants, they were assigned a participant identification code (e.g., Participant 01, Participant 02).

To ensure rigor, the four criteria for trustworthiness as described by Guba and Lincoln including credibility, dependability, transferability, and confirmability were used [20]. Prolonged engagement with the analysis was conducted to promote credibility, including comparing outcomes to the existing literature once analysis was completed. To promote dependability, rich descriptions of the data were created and all co-authors audited the findings [21]. Data saturation and detailed descriptions of the results enhanced potential for transferability to other community settings and contexts. To achieve confirmability, the researcher sought to ensure that all the stages of analysis were explained and that the results were presented in sufficient detail. The research team took time to identify preconceived ideas and assumptions and to reflect on their own positions. Reflexive journaling was undertaken to document thoughts and feelings about the research process and to identify ideas about the analysis as they came about. Furthermore, team members met to discuss emerging thematic analysis and to review any discrepancies or areas that were unclear. However, transcripts were not provided to the participants for their input, comments, or corrections.

### 2.3. Ethical Considerations

Harmonized Research Ethics Board approval was obtained from the University of Northern British Columbia and the Northern Health Authority prior to data collection (H20-00581-A002). Participation in this study was voluntary. Informed consent was obtained from all subjects involved in the study prior to participation in the study.

## 3. Results

Eleven RNs participated in this study (mean age 52.5 years). Thirty-six percent of the participants had a Diploma in Nursing (*n* = 4), 27% had a Bachelor’s in Nursing (*n* = 3), and 36% had a Master’s in Nursing (*n* = 4). Despite attempts to generate a diverse sample, all participants reported female sex and gender (See Table 1). Participants’ overall work experience as a RN averaged 24.6 years (range 2~40 years), with a mean average of 19.5 years of work experience in rural communities. The job positions and roles reported by participants were diverse and included: community nurse, home and community care clinical educator, home support field supervisor, case manager, pediatric nurse, and long-term care facility nurse.

Participants described the value current and future technologies play to support the ability of RNs to do their jobs in rural and northern communities and expressed the opportunities existing technologies provide to support provision of high-quality patient centered care. As one of the participants mentioned, “Technology for me really plays a huge role in delivering the best care” (Participant 004). Participants shared their experiences on the gradual introduction of technologies and increased integration into daily work routines. Participant (002) noted “it’s so integrated with what I do, so, like when I say, you know, how does it like it’s not even a support it’s integral”. Experiences during the COVID-19 pandemic accelerated the importance of virtual technology.

During the interviews, participants shared how viewing the video showing an individual with essential tremors demonstrating use of the Steadiwear antivibration glove was realistic and reflective of individuals they care for as part of their clinical practice. All participants reported that use of the Steadiwear antivibration glove as a tool to enhance independence of the person with essential tremors would in turn subsequently reduce the need for health support. Perceived positive impacts on the independence of persons with essential tremors included enhancing safety, enabling independence in personal hygiene, improving communication and social interactions, increasing quality of life, reducing admission to long-term care facilities along with increasing in self-esteem and self-confidence (Figure 1). The increased independence was also predicted to reduce potential workloads among both rural nurses and personal care workers. “I think it would make some more independent so would be a little bit less reliant on their support” (Participant 006). Use of the Steadiwear antivibration glove also showed potential to enhance the ability of persons to be more independent for a longer period. One participant explained: “I think that it would decrease the time that he would meet with people that could assist him I think it again would give him that independency to a great extent” (Participant 008). 

Participants noted that the Steadiwear antivibration glove showed great potential to influence individuals’ ability to remain independent across a number of ADL and IADL tasks, including the ability to take medication independently, prepare meals, and independently consume food and drinks (Figure 2 and Figure 3). Participant 8 noted “I think this will be a positive impact that aspect of life, because they’ll be able to do the daily living activities by themselves in a better way. Like taking water from the fridge, drinking water, or making food by themselves or preparing snacks for them”. Furthermore, participant 5 shared “if he was on medications, it would help him to be able to take them on his own”. 

Participants highlighted how use of the Steadiwear antivibration glove could potentially increase abilities to engage in personal hygiene and self-care activities. “They will be able to work on their hygiene care and everything because they feel confident that they can perform, they can do, they are able to do their basic needs” (Participant 007). This included grooming, dressing, and toileting. One participant commented: “They’d be able to go to the washroom independently” (Participant 006). Participants further described how use of the Steadiwear antivibration glove could enable the user to engage in IADLs, including banking, shopping, and social activities which required steadiness and dexterity. As one participant commented:
It’s obvious that he would be more independent in banking, and using the phone, in writing a shopping list, and using a computer, in using a cell phone, and being able to manage a microwave.(Participant 011)

Enhanced safety was emphasized in the ability of individuals to perform activities more independently and safely, such as being able to safely cook and to drink hot fluids without spilling. Participant 6 noted that an individual using the Steadiwear antivibration glove “Probably will have less burns from dropping hot drinks”. Furthermore, participant 11 emphasized that even though the individual may still need help or supervision when cutting things, the user would be able to use a knife more independently than before. “He would be able to cut his food whereas prior to using the glove I can’t see that he’d be able to successfully cut his food”.

The participants also reflected on the potential improvements in patient independence and ability to age in place longer because of the use of this technology. One participant commented “I think that it also helps with safety that the need for admission to long-term care facilities may decrease” (Participant 010). This kind of technology can potentially help people or support people in living longer at home. Furthermore, the Steadiwear antivibration glove showed potential to help persons with essential tremor to age in place. For example, one participant reported:
Yeah, and I think it would increase his ability to stay home longer. Right. So, our goal is to keep people at home. And, you know, something like that may help increase visibility to stay home, especially if it increases his willingness to go out and socialize.(Participant 005)

Other participants highlighted further that use of the Steadiwear antivibration glove could support persons with essential tremor wishing to age in place in their home setting by preventing or delaying the need for admission to a long-term care facility. For example, participant 9 shared that “It [Steadiwear antivibration glove] gives him that independence again, even in the home, I feel he would be able to stay at home”. This was echoed by participant 8 who also commented: “I think that [use of the Steadiwear antivibration glove] would help prevent admission, early admission” and participant 5, who noted “It would likely delay their entrance into long term care because they could maintain some independence at home in the community, and the level of independence may depend on the resources you have locally”.

Participants connected the increase in the person’s independence through use of the Steadiwear antivibration glove to an increase in the individuals’ quality of life and well-being, describing how increases in independence could lead to confidence and subsequently to motivation to connect with others. Participants shared how many persons with essential tremor who may have withdrawn from social activities due to lack of confidence, shame, and stigma could potentially resume their social interactions with use of the Steadiwear antivibration glove. Participant ten noted that “Then they will try to involve in social activities where they can sit with others, and they can manage their own task” Furthermore, participant 5 noted that with use of the Steadiwear antivibration glove, individuals may be less self-conscious and able to engage in public activities such as eating in a restaurant. Participant 5 shared “He can probably use a regular glass; I would imagine on the shakes like that would have adaptive utensils”. Further, participant 3 noted how with use of the Steadiwear antivibration glove “They can go and eat a bowl of cereal by themselves when they’re hungry”. Participants described how people with essential tremors can experience challenges going outside of their home. For example, participant 5 described how they “would imagine that that gentleman [person experiencing essential tremors] would have very difficult time even opening a door” and yet with the use of the Steadiwear antivibration glove, this same participant optimistically noted that “He might even go out to a restaurant right so he’s going to be more comfortable going out in a public” (Participant 005). 

Another key perceived positive effect of using the Steadiwear antivibration glove was seen as increasing the person’s self-esteem and self-confidence. “They [the gloves] will improve their activity, but also their self-esteem and confidence. So, they are able to perform the daily living tasks” (Participant 001). Another effect of increasing independence can be increasing self-confidence. “So, I said to have them, not to rely on people, you know, for performance of their ADL, to be able to do things independently and in a way that promote that sense of self-esteem” (Participant 004). Participants then further described their own clinical observations, noting an association between an increase in the person’s self-confidence and increase in self-esteem with an increase in an individual’s willingness to engage with others and increase social interactions. “So, I feel that these are some of the things that that will increase their self-esteem and self-confidence and their social engagement as well” (Participant 001). Participants then connected the increased abilities and self-confidence to a reduced need for care and support from others.
It can definitely lessen the burden of care if he likes to use that kind of phrasing. For those who are around him. Both those who live with him if he lives with others, or even a nurse or an LPN.(Participant 001)

Participants shared important concerns and potential limitations to introducing the Steadiwear antivibration glove in rural community-based settings. Concerns included the availability of educational materials and support for care providers to teach their clients how to use the technology as well as capacity and resources to safely maintain the technology.
Some of the things that are being implemented might be developed for larger hospitals or larger settings, and it doesn’t necessarily fit for what they’re doing in rural areas… It was a lot of things we need to learn in his care which was okay, we were ready to do that… But it was super scary for the nurses because they had to make sure that, you know, have batteries for it and would change it from regular basis because the guy couldn’t. You know, if they could not do that for him so like life altering thing if that process is messed up. Anyways, as awesome as this was for him and his family. If it broke down. There was nobody on the island who could fix it. There was nobody. The technology was so advanced. It was a huge problem.(Participant 010)

One criteria that decision-makers need to consider regarding the effects that technologies such as the Steadiwear antivibration glove may have on RNs workload was the degree of information needed to introduce and sustain use of the Steadiwear antivibration glove among persons with essential tremors. 

## 4. Discussion

The analysis of the study findings revealed that rural RNs had a positive attitude towards technology use across diverse health care settings, recognizing the potential for technology like the Steadiwear antivibration glove to support RNs to deliver higher quality patient care. In line with previous research [22,23,24], RNs in the current study recognized that barriers to technology use of the Steadiwear antivibration glove could be overcome with the right training, proper decision making, and appropriate infrastructure. 

RNs were enthusiastic about the potential use of the Steadiwear antivibration glove, as an example of a new wearable technology to support their clients’ independence. They identified some possible positive effects of using the Steadiwear antivibration glove on the quality of life of persons living with essential tremor, including the potential for increased social interactions, independence in ADLs and IADLs, and delayed need for transition to a LTCF. Increasing the independence of persons living with essential tremors by using the Steadiwear antivibration glove can also improve their personal hygiene and increase their self-esteem and self-confidence [25,26,27]. Like other technologies, users of the Steadiwear antivibration glove may face barriers to access the technology including cost and product availability, as well as in support and training to adapt to a new technology [28,29,30].

Before implementing any technology, the needs of the work environment should be assessed. Therefore, the needs and capacity of both people living with essential tremor and their healthcare providers should be considered. In rural and northern areas, it is important to examine factors to support education on how to use, access, and safely maintain the technology, as well as how to ensure that technologies to support independence of persons with essential tremors are affordable and available [31,32]. 

Implementing advanced education and training focused on new and emerging technologies has been shown to increase nurses’ knowledge of how to use technologies properly, which is necessary to help reduce workload demands [32]. However, when technology does not function properly or RNs do not possess the necessary skills to utilize it, despite its potential to reduce workload, it may fail to do so and can ultimately result in an increased workload. Registered Nurses may benefit from expanded opportunities to learn about how new technologies, such as the Steadiwear antivibration glove, to better understand how its use may support clients they serve in their practice settings. Registered Nurses who are not confident about using technology should be supported to increase their confidence and competence using the technologies in routine care settings. Consistent training and support are necessary to ensure RNs are prepared to support clients to use the technology [30]. When nurses have the confidence and enough knowledge to use technology, their workload may indirectly reduce [33].

Despite attempts to attract a diverse pool of participants, a limitation to the current study was the absence of male and gender diverse participants. However, due to the small sample size, this was expected as most practicing RNS are female. Further research should expand the diversity of the RNs sampled to obtain more comprehensive and generalizable data, male and gender diverse participants must also be included in future studies. Furthermore, the research was undertaken in one regional area. Engaging more diverse rural communities may yield new insights.

## 5. Conclusions

The Steadiwear antivibration glove was perceived as an important technology to support high quality patient care for persons living with an essential tremor. It was notable that the Steadiwear antivibration glove showed the potential to reduce the workload of healthcare providers by enhancing the ability of persons living with essential tremors to remain independent in many ADL and IADL abilities. To ensure the successful implementation of the Steadiwear antivibration glove, it is important to address key barriers, including client knowledge in how to access and use the Steadiwear antivibration glove, as well as knowledge, awareness, and management of the use of the glove by family, friends, and healthcare providers. 

The ability for persons living with Essential Tremor to age in place and remain independent in their activities of daily living and instrumental activities of daily living can be quite challenging. Especially in rural communities, health care services and home care support can be greatly limited resulting in premature admissions to long-term care support. As new technologies show great promise to support persons to live independently, we sought to examine whether use of the Steadiwear antivibration glove would be perceived as support to not only enhance the independence of the user but also to help reduce demands for more intensive levels of care support. Findings from this study clearly suggest that the Steadiwear antivibration glove can enhance the users ADL and IADL abilities thereby reducing their need for LTC and supporting them to age in place in the community longer. We also highlight that to achieve the benefits to support users to age in place, it is critical to address key barriers including supporting users to access and use the Steadiwear antivibration glove and to enhance knowledge, awareness, and management of use of the glove by family, friends, and healthcare providers.

## Figures and Tables

**Figure 1 ijerph-21-00714-f001:**
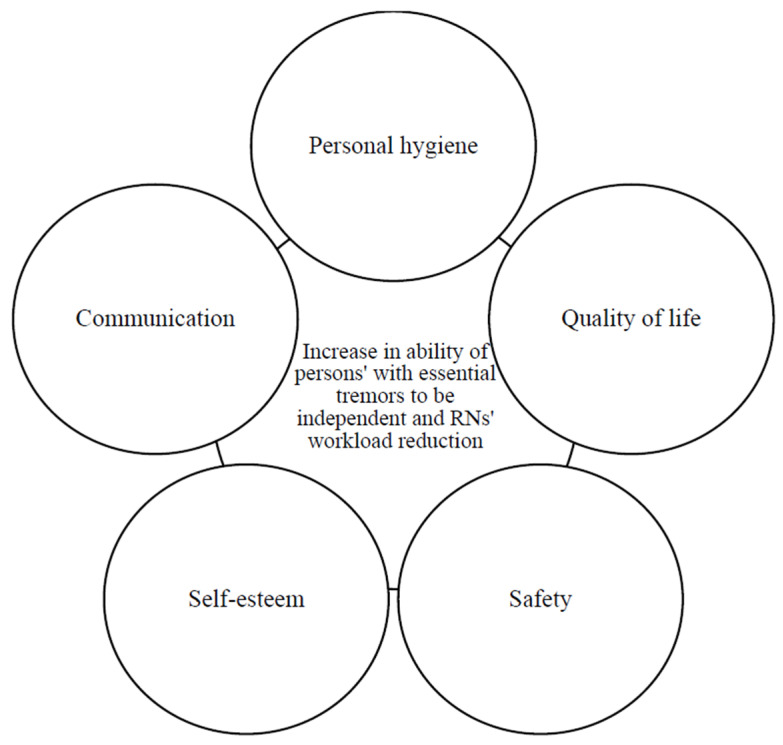
Key factors associated with use of Steadiwear antivibration glove by persons with essential tremor.

**Figure 2 ijerph-21-00714-f002:**
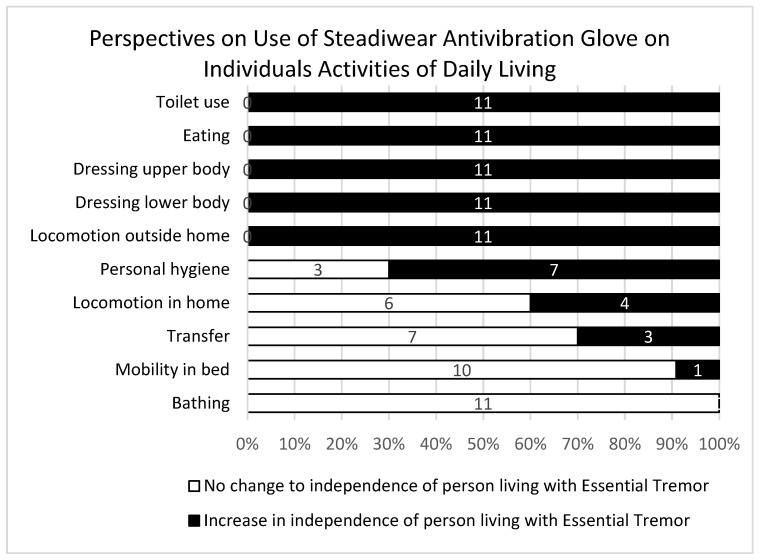
Participants’ perspectives of the activities of daily living of a person living with essential tremors while using the Steadiwear antivibration glove in comparison to prior use of the glove (*N* = 11).

**Figure 3 ijerph-21-00714-f003:**
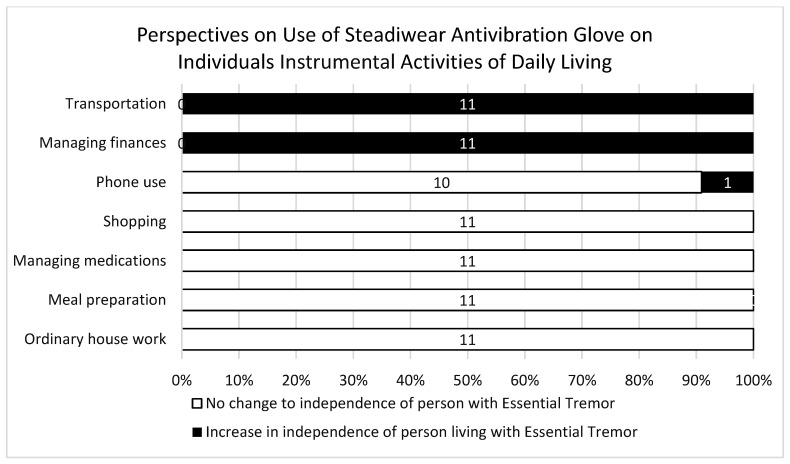
Participants’ perspectives of instrumental activities of the daily living of a person living with essential tremors while using the Steadiwear antivibration glove in comparison to prior use of the glove (*N* = 11).

**Table 1 ijerph-21-00714-t001:** Demographics of the participants.

Demographic Variables	Percentage (*N* = 11)
Age	
20–30	9.0 (1)
30–40	0 (0)
40–50	36.4 (4)
50–60	36.4 (4)
60–70	18.2 (2)
Gender	
Female	100 (11)
Male	0 (0)
Sex	
Female	100 (11)
Male	0 (0)
Education Level	
Diploma	36.4 (4)
Bachelor’s degree	27.3 (3)
Master’s degree	36.4 (4)

## Data Availability

Data may be made available upon request to the corresponding author.

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
