# Peer review of "Use of Antivibration Technology to Reduce Demands for In-Home Nursing Care and Support in Rural Settings for Persons with Essential Tremors: A Qualitative Study"

_ijerph, 2024, doi:10.3390/ijerph21060714_

Round 1

Reviewer 1 Report

Comments and Suggestions for Authors

Dear Author,

This article's theme is of great importance for nurses and patients in the context of Community health. However, it presents some weaknesses that I highlight below:

1. Abstract: the type of study is not identified in the Methods item. Must reformulate;

2. Keywords, it is not clear why the key word “aging in place, interRAI, homecare”, what does this word add to the knowledge of the paper? On the other hand, shouldn't there be keywords associated with technology? It must be reformulated considering the DeCS/MeSH Health descriptors.

3. Methods- should better explain the type of study. It is an understatement to mention that this study is a qualitative study. It should better classify the type of study; Does not present the inclusion and exclusion criteria in the study - They must explain

They should better describe the data collection procedure associated with the interview, namely when, where and how it was carried out

4. In line 112 they state that “Following review of the video, participants reflected hypothetically on the usability 112 of the Steadi-One glove noting” how was this reflection made? How long does it take between watching the video and the interview? It must be explained

5. Results: Figure 2 is captioned “participants Perspectives on ADL and IADL Scale of a Person Living with essential tremors while. What scales are these? Why didn't they mention them in the methods?

6. Discussion, in line 318 they state that “To ensure successful implementation of the Steadi-one glove it is important 366 to address key including barriers client knowledge in how to access and use the Steadi” What barriers are these? How did you identify these barriers? – It must be explained

7. In line 340 they state that “Implementing advanced education and training focused on new and emerging tech-340 nologies has been shown to increase nurses’ knowledge of how to use technologies 341 properly which is necessary to help reduce workload demands. “How did you come to this conclusion? They must explain

Best regards,

Author Response

REVIEWER 1

RESPONSE TO REVIEWERS

1. Abstract: the type of study is not identified in the Methods item. Must reformulate;

Thank you for this suggestion. We have added the study design to both the title and the abstract.

2. Keywords, it is not clear why the key word “aging in place, interRAI, homecare”, what does this word add to the knowledge of the paper? On the other hand, shouldn't there be keywords associated with technology? It must be reformulated considering the DeCS/MeSH Health descriptors.

We appreciate the opportunity to provide clarity here. interRAI is a relevant keyword as it relates to the standardized community assessments used both as the foundation in the study.  Further, “aging in place” is relevant as this technology could be seen to help support individuals to age in place.  This is a very common term in Canada. We appreciate the reviewers comment however and have now added have added the term “Independent Living” which is the MeSH term for “Aging in Place.”

3. Methods- should better explain the type of study. It is an understatement to mention that this study is a qualitative study. It should better classify the type of study; Does not present the inclusion and exclusion criteria in the study - They must explain

They should better describe the data collection procedure associated with the interview, namely when, where and how it was carried out

We appreciate this suggestion and have added to the methods section to provide clarity and further detail as requested.

4. In line 112 they state that “Following review of the video, participants reflected hypothetically on the usability 112 of the Steadi-One glove noting” how was this reflection made? How long does it take between watching the video and the interview? It must be explained

We have noted in the manuscript that the video was shown at the beginning of the interview.  As it was part of the interview, it was fresh in the minds of the participants when they were providing their feedback.

5. Results: Figure 2 is captioned “participants Perspectives on ADL and IADL Scale of a Person Living with essential tremors while. What scales are these? Why didn't they mention them in the methods?

Thank you for this suggestion. We have written out ADL and IADL in the title.  The names on the bars were the categories we inquired about.

6. Discussion, in line 318 they state that “To ensure successful implementation of the Steadi-one glove it is important 366 to address key including barriers client knowledge in how to access and use the Steadi” What barriers are these? How did you identify these barriers? – It must be explained

These barriers were described by participants and are noted in lines 318 -342.

7. In line 340 they state that “Implementing advanced education and training focused on new and emerging technologies has been shown to increase nurses’ knowledge of how to use technologies properly which is necessary to help reduce workload demands. “How did you come to this conclusion? They must explain

Thank you for this note. We have added references.

REVIEWER 2

RESPONSE TO REVIEWER

REVIEW: The introduction provides sufficient information and includes relevant references. However, it requires an extensive review of references, as most references are more than five years old. Thus, as noted in the rest of the text.

We have updated the introduction to provide more references and greater detail.

Considering a qualitative study, the article was reviewed based on the propositions of the Consolidated Criteria for Reporting Qualitative Research (COREQ), which has been widely used to report qualitative research. In this aspect, some points of the study method should be better described:

1) It requires expanding the characterization of the research team and the researcher's reflexivity.

2) There is a lack of methodological guidance to support the study

We thank the reviewer for this comment.  We have added further detail to the methods section. And also have attached the COREQ as requested.

3) Regarding sampling, I suggest pointing out how the study scenario was defined. What is the relationship between the researchers and the study scenario. I suggest clarifying whether there was an inclusion criterion for the inclusion of participants, considering time working in rural communities. Describe how participants were approached? Was there a refusal to participate? And what reason?

Thank you for this comment. We have added further detail to describe the sampling in the methods section to address the reviewers’ questions.

4) I suggest describing the context of the interview scenario, considering the use of digital media. Was there any interference? How were the interviews scheduled?

We have added further detail on this in the methods section.

5) It is not clear about the use of an interview guide, I suggest expanding the use of the guide, how it was structured and whether there was a pilot test.

We have added further detail on this in the methods section.

6) Regarding the interviews, were they recorded? How long do the interviews last?

We have added further detail on this in the methods section.

7) Were transcripts returned to participants for comments or corrections?

We have added further detail on this in the methods section and noted this as a limitation of the study as we were not able to return the transcripts to the participants.

8) Regarding data analysis in the method: it was not clear as the data coders, as well as the derivation of themes, were not described. Despite the use of supplementary file 1.

The authors involved in the data coding and analysis is noted at the end of the manuscript in the authors contributions.

9) Was software used to manage data?

Data was transcribed into Microsoft word.  Not analysis software was used.

10) I suggest describing the coding of participants in the method.

We have added further detail on the coding in the methods section.

REVIEWER 3

RESPONSE TO REVIEWER

- the references and the data aren't recent. In the background missing specfic evidence of istrument use or potential of RN in rural area (if present...);

We have updated the references and added further information to the background section.

- the editorial process is correct how in line 42, 44 (and for the rest of the text) the citation is after the point?...

We have edited this section as requested.

- in the limitation I think can be added the pre-existing professional experience of the RN enrolled (the gender difference indicated is frail...). If is possible in the Table 1 I added this part of declared in the limitations;

We have noted this as a limitation on page 10.

- in the Figure 3 (indicated in the text how second figure 2...) there is typing error ("mnaging"...);

We have addressed this typo as noted.

- in the references (that I suggest to improved) there are part underlines...

We have updated the references.

Reviewer 2 Report

Comments and Suggestions for Authors

ABSTRACT: Article aimed to examine Registered Nurses (RN) perceptions of the potential of the Steadi-One glove to reduce the need for in-person support from community healthcare professionals. Topic of relevance to the scientific field and health practice, considering the growing integration of technologies in the health sector.

REVIEW: The introduction provides sufficient information and includes relevant references. However, it requires an extensive review of references, as most references are more than five years old. Thus, as noted in the rest of the text.

Considering a qualitative study, the article was reviewed based on the propositions of the Consolidated Criteria for Reporting Qualitative Research (COREQ), which has been widely used to report qualitative research. In this aspect, some points of the study method should be better described:

1) It requires expanding the characterization of the research team and the researcher's reflexivity.

2) There is a lack of methodological guidance to support the study.

3) Regarding sampling, I suggest pointing out how the study scenario was defined. What is the relationship between the researchers and the study scenario. I suggest clarifying whether there was an inclusion criterion for the inclusion of participants, considering time working in rural communities. Describe how participants were approached? Was there a refusal to participate? And what reason?

4) I suggest describing the context of the interview scenario, considering the use of digital media. Was there any interference? How were the interviews scheduled?

5) It is not clear about the use of an interview guide, I suggest expanding the use of the guide, how it was structured and whether there was a pilot test.

6) Regarding the interviews, were they recorded? How long do the interviews last?

7) Were transcripts returned to participants for comments or corrections?

8) Regarding data analysis in the method: it was not clear as the data coders, as well as the derivation of themes, were not described. Despite the use of supplementary file 1.

9) Was software used to manage data?

10) I suggest describing the coding of participants in the method.

The research question is original and well defined. However, acceptance of the manuscript would depend on important revisions. The author needs to provide a point-by-point response or provide a rebuttal if some of the reviewer's comments, particularly on the method and results. Requires inclusion of current references, in their entirety the articles are more than five years old. Data and analysis requires review to present appropriately.

Author Response

(The authors gave the same response as above.)

Reviewer 3 Report

Comments and Suggestions for Authors

Dear Authors,

good work but I modest think will be improved;

- the references and the data aren't recent. In the background missing specfic evidence of istrument use or potential of RN in rural area (if present...);

- the editorial process is correct how in line 42, 44 (and for the rest of the text) the citation is after the point?...

- in the limitation I think can be added the pre-existing professional experience of the RN enrolled (the gender difference indicated is frail...). If is possible in the Table 1 I added this part of declared in the limitations;

- in the Figure 3 (indicated in the text how second figure 2...) there is typing error ("mnaging"...);

- in the references (that I suggest to improved) there are part underlines...

Good work.

Best Regards

Author Response

(The authors gave the same response as above.)

Round 2

Reviewer 2 Report

Comments and Suggestions for Authors

GENERAL REVIEW: The introduction still calls for an extensive review of references, as most of the references are more than five years old. Thus, as observed in the rest of the text and especially the discussion. Only 2 references have been updated in relation to the first revision, despite the fact that the change in references was indicated in the text.

The article has been revised on the basis of previous propositions; some points of the study method still need to be better described:

1) Regarding recruitment, is it still unclear how the participants were approached? Were there any refusals to participate? And for what reason?

2) Check the use of the term "exploratory interview" line 131. Isn't exploratory a type of study?

3) You still haven't presented the characterization of the research team and the reflexivity of the researcher.

4) With regard to sampling, it has not yet been pointed out how the study setting and the participants in this setting were defined. What is the relationship between the researchers and the study scenario?

5) There is still a need to describe the coding of the participants in the method (Participant 001...).

Author Response

REVIEWER 3

RESPONSE TO REVIEWER

GENERAL REVIEW: The introduction still calls for an extensive review of references, as most of the references are more than five years old. Thus, as observed in the rest of the text and especially the discussion. Only 2 references have been updated in relation to the first revision, despite the fact that the change in references was indicated in the text.

We have further updated and increased the number of references published more recently in the introduction section.

1) Regarding recruitment, is it still unclear how the participants were approached? Were there any refusals to participate? And for what reason?

We have added further detail to this in section 2.1.

2) Check the use of the term "exploratory interview" line 131. Isn't exploratory a type of study?

We have removed the word ‘exploratory’ to ensure clarity.

3) You still haven't presented the characterization of the research team and the reflexivity of the researcher.

Details on reflexivity may be found in the third paragraph of section 2.1.

4) With regard to sampling, it has not yet been pointed out how the study setting and the participants in this setting were defined. What is the relationship between the researchers and the study scenario?

With regard to sampling, we note this in section 2.1 that “To be eligible to participate, participants needed to be RNs with experience working in a rural community (defined as a community with less than 10,000 residents) [12] and in community health settings, including home and community care, community case managers, and long-term care nursing”  

There was no pre-existing relationship between the researchers and the study scenario.

5) There is still a need to describe the coding of the participants in the method (Participant 001...).

We provide detail to this in second paragraph of section 2.2. and have added an additional sentence for further clarity.